# Addressing Noise and Estimating Uncertainty in Biomedical Data through the Exploration of Chemical Space

**DOI:** 10.3390/ijms232112975

**Published:** 2022-10-26

**Authors:** Enrique J. deAndrés-Galiana, Juan Luis Fernández-Martínez, Lucas Fernández-Brillet, Ana Cernea, Andrzej Kloczkowski

**Affiliations:** 1Group of Inverse Problems, Optimization and Machine Learning, Department of Mathematics, University of Oviedo, C/Federico García Lorca, 18, 33007 Oviedo, Spain; 2Department of Informatics and Computer Science, University of Oviedo, C/Federico García Lorca, 18, 33007 Oviedo, Spain; 3DeepBioInsights, C/Federico García Lorca, 18, 33007 Oviedo, Spain; 4The Steve and Cindy Rasmussen Institute for Genomic Medicine, Nationwide Children Hospital, Columbus, OH 43205, USA; 5Department of Pediatrics, The Ohio State University, Columbus, OH 43205, USA

**Keywords:** drug design, drug discovery, phenotype prediction, artificial intelligence, noise and uncertainty

## Abstract

Noise is a basic ingredient in data, since observed data are always contaminated by unwanted deviations, i.e., noise, which, in the case of overdetermined systems (with more data than model parameters), cause the corresponding linear system of equations to have an imperfect solution. In addition, in the case of highly underdetermined parameterization, noise can be absorbed by the model, generating spurious solutions. This is a very undesirable situation that might lead to incorrect conclusions. We presented mathematical formalism based on the inverse problem theory combined with artificial intelligence methodologies to perform an enhanced sampling of noisy biomedical data to improve the finding of meaningful solutions. Random sampling methods fail for high-dimensional biomedical problems. Sampling methods such as smart model parameterizations, forward surrogates, and parallel computing are better suited for such problems. We applied these methods to several important biomedical problems, such as phenotype prediction and a problem related to predicting the effects of protein mutations, i.e., if a given single residue mutation is neutral or deleterious, causing a disease. We also applied these methods to de novo drug discovery and drug repositioning (repurposing) through the enhanced exploration of huge chemical space. The purpose of these novel methods that address the problem of noise and uncertainty in biomedical data is to find new therapeutic solutions, perform drug repurposing, and accelerate and optimize drug discovery, thus reestablishing homeostasis. Finding the right target, the right compound, and the right patient are the three bottlenecks to running successful clinical trials from the correct analysis of preclinical models. Artificial intelligence can provide a solution to these problems, considering that the character of the data restricts the quality of the prediction, as in any modeling procedure in data analysis. The use of simple and plain methodologies is crucial to tackling these important and challenging problems, particularly drug repositioning/repurposing in rare diseases.

## Highlights

Uncertainty is intrinsic to any estimation problem and mainly comes from noise in data and modeling hypotheses. Uncertainty is introduced by the question that we try to elucidate.Noise in data always introduces errors in the predictions because it enters the cost function in a nonlinear way. Uncertainty also means ambiguity in the identification. The best way to address both issues is sampling.Sampling in high-dimensional spaces is a very challenging task that requires the use of dimension-reduction techniques. When the character of a problem is determined, it becomes linearly separable.Sampling is hampered by high computational costs to perform forward predictions. Random sampling methods are highly ineffective. Smart model parameterizations, forward surrogates, and parallel computing are sampling techniques that can overcome these bottlenecks.

## 1. Introduction

### The Concept of Noise and Its Role on Uncertainty in Biomedical Data

In signal processing, noise is any undesirable modification affecting a signal in any of the steps concerning its acquisition and/or post/preprocessing. Noise is inherent to any process of measurement since it is practically impossible in real systems to acquire data without noise. Noise in data can be of different types depending on its power spectrum, which is calculated through Fourier analysis of the signal x(t), considered a temporal series. For a finite-energy signal x(t), its energy can be calculated as follows:(1)$$E=\int_{-\infty }^{+\infty }{\left|x\left(t\right)\right|}^{2}dt.$$

The density of the spectra describes how the energy E is distributed in the frequency domain. For that purpose, the Fourier transform of the signal is defined as:(2)$$F\left(x\left(t\right)\right)=X\left(f\right)=\int_{-\infty }^{+\infty }e^{-2\pi ift}x\left(t\right)dt,$$
where f is the frequency. Expression (2) measures the similarity between the complex exponential $e^{-2\pi ift}$ at the frequency f and the signal x(t). Moreover, one of the main properties of Fourier analysis is Parseval’s theorem, which states the conservation of energy when passing from the time domain to the frequency domain, that is:(3)$$\int_{-\infty }^{+\infty }{\left|x\left(t\right)\right|}^{2}dt=\int_{-\infty }^{+\infty }{\left|X\left(f\right)\right|}^{2}df.$$

Expression (3) states that there is no loss of energy in the frequency analysis; in other words, the information in the signal x(t) is also contained in its spectrum |X(f)|. When measurement noise δx(t) is added to the signal, the power spectrum of the perturbed signal is obviously the sum of the original spectrum of the signal and that of the noise. White noise has a power spectrum, which is flat; that is, the noise has equal power in all frequencies. When the power spectrum of the noise does not follow this definition, the noise is categorized by color, depending on how its power spectral density relies on $1/f^{\beta }$, with *β* = 0 for white noise, 1 for pink noise, 2 for red (Brownian) noise, and *β* > 2 for black noise. The aim of this explanation is to provide some insights into how noise contaminates the signal and makes its interpretation tough.

In fact, signal processing consists of designing special types of filters called linear time-invariant (LTI) systems S(t), which act on the original signal through the convolution operator, denoted by ∗:(4)$$y\left(t\right)=S\left(x\left(t\right)\right)=S\left(t\right)*x\left(t\right)=\int_{-\infty }^{+\infty }S\left(\tau \right)x\left(t-\tau \right)d\tau .$$

This type of system is very useful because the convolution in the spectral domain transforms to a multiplication, that is:(5)Y(f)=S(f)·x(f).

Therefore, calling Y(f) the spectrum of the signal without noise, the LTI system that performs this task can be calculated as the inverse Fourier transform of the spectrum S(f):(6)$$S\left(t\right)=F^{-1}\left(\frac{Y\left(f\right)}{x\left(f\right)}\right).$$

Briefly, these are the basics of the digital processing applied to noise attenuation in signals. However, in computational biology and in drug design, the problem is a little bit more complicated than in signal filtering and involves the theory of inverse problems, i.e., the part of mathematical modeling that relates causes and observed effects. This is also called reverse engineering or system identification. Some cases that are discussed in this paper are:The phenotype prediction problem, which consists of identifying the altered genetic pathways that are responsible for the development of a disease, given a set of samples belonging to both disease and healthy control classes.The protein folding problem, i.e., protein tertiary structure prediction from its sequence.The single-nucleotide polymorphisms (SNPs) problem, which consists of predicting if a given mutation (or a set of mutations) in the genome correlates with disease. Mutations are deleterious if they decrease the fitness of an organism and might be causing a disease. Mutational effects can be favorable, harmful, or neutral, depending on their context or position. The deleterious mutation hypothesis assumes that sex exists to purge a species of damaging genetic mutations. The majority of deleterious mutations are marginally deleterious, and the introduction of each mutation has an increasingly considerable effect on the fitness of the organism [1]. Most non-neutral mutations are deleterious. A related problem consists of predicting how a given mutation in the DNA affects the expression of the different genes in the transcriptome.The de novo drug design problem, which consists of optimizing the structure of new drugs with the mechanism of action designed to fight a disease, maximizing efficacy, and minimizing harmful side effects.

## 2. Discussion

A very popular phrase of Benjamin Franklin is that *“nothing is certain except death and taxes“*. Uncertainty is intrinsic to any estimation problem and mainly comes from noise in data and modeling hypotheses. Noise in data always introduces errors in the predictions because it enters the construction cost function, and the inverse problems consist of identifying the possible causes from a discrete sampling of the effects. Uncertainty means ambiguity in this identification process; that is, there might exist different plausible scenarios providing the same effect.

It is important to understand that in a parameter identification problem, sometimes, the biggest uncertainty is introduced by the question that we try to elucidate. Let us provide an example. Imagine that in a preclinical trial we try to predict the effect of a given drug on the expression of a given gene, relating this change in expression to its chemical structure. In this case, this is the hardest hypothesis, because it might not be the case. Once the identification problem is designed, how the answer is parametrized is crucial. Trying to predict exactly the change in expression induced by this drug is not the same as deciding if the effect is overexpression, underexpression, or neutral. Therefore, modeling should be contemplated as an art. Inverse and parameter identification problems are very challenging because they are ill-posed; that is, they usually lack a unique, stable solution.

The only way to address both issues (noise in data and uncertainty) is by sampling the different scenarios and deciding accordingly. Nevertheless, sampling in high-dimensional spaces is a very challenging task that requires the use of dimension-reduction techniques. When the alphabet of a problem is found, it becomes linearly separable. Sampling is hampered by high computational costs to perform forward predictions. Random sampling methodologies are obviously forbidden. Smart model parameterizations, forward surrogates, and parallel computing are needed to overcome these bottlenecks.

In our opinion, the key findings in the research performed in this field so far come from the side of the algorithms provided by the applied mathematics and computer science community. The main weakness comes from the data side, since not enough good-quality open databases are available to find the correct answers. We believe that the ultimate goal is to put together different expert teams able to simultaneously understand machine learning and artificial intelligence techniques, with a deep understanding of genomics and drug design. This is not an easy challenge, because this kind of problem cannot be solved with a black-box type model. Moreover, big data is always little from the side of the samples. The pharma industry and hospitals should establish more collaborative and win–win approaches to boost the use of this powerful technique in the design of new cures, simultaneously impacting costs and time. This, for us, is clearly the future, and it is happening now.

## 3. Methodology

### 3.1. Inverse Problem

An inverse problem in discrete form consists of a set of equations:(7)F(m)=dobs,
where m∈ℝn is an *n*-dimensional vector of model parameters that should be identified. Here F, a vector field, is the forward mathematical model, and **d^obs^** is an *s*-dimensional vector of observed data (or effects).

The forward problem (or prediction problem) consists of calculating F(m); that is, knowing the cause m, we predict the effect $d^{pre}=F\left(m\right)$. The superscript *pre* in **d** means that $d^{pre}$ is the result of a numerical prediction. It follows that it is necessary to measure the distance between the prediction and the observation. For that purpose, the data space containing the measurable data and the predictions should have a kind of metric $\|F\left(m\right)-d^{\mathrm{obs}}\|{}_{p}$ to measure the distance between observations and predictions and to estimate how good they are. The inverse problem is based on finding the origin, **m**, and knowing the theoretical model **F** (or the forward model) to fit the observed data, **d^obs^**.

The main difference between the forward and the inverse problems concerns their well-posedness, i.e., the well-posed character of the solution. In fact, while the direct problem has a unique and stable solution (it is well-posed), the inverse problem either has no solution, the solution is not unique, or it does not only depend on the data, and small perturbations in the data due to the noise can cause drastic (and erroneous) modifications of the identified model **m**.

Concerning the ill-posed character of the inverse problem, it is important to determine the uncertainty space, which is composed of the set of models that fit the data within the same error bounds $E_{tol}$:(8)$$M_{tol}=\left\{m:\;C\left(m\right)=\frac{\|F\left(m\right)-d^{obs}\|{}_{2}}{\|d^{obs}\|{}_{2}}<E_{tol}\right\}$$

In this case, to define $M_{tol}$, we use the Euclidean norm, but other norms can be used.

Fernández-Martínez et al. [2] found that the topography of the cost function in linear inverse problems (when F corresponds to a linear operator between m and $d^{obs}$) corresponds to the set of models inside a linear hyperquadric whose center is the least-squares solution and whose axes are linked to the eigenvalues and eigenvectors of the quadric matrix $F^{T}F$. Geometrically, this corresponds to a flat elongated valley with null gradients. The case of linear systems corresponds to finding the solution of the linear system $Fm=d^{\mathrm{obs}},$ where F is a rectangular matrix, whose rows are observed data ($d^{\mathrm{obs}})$ and whose columns depend on the size of m. In the case of purely overdetermined linear systems, the noise in data perturbs the least-squares solution:(9)$$F^{T}Fm=F^{T}d^{\mathrm{obs}}$$
and might originate spurious solutions, depending on the ill-conditioning of F; thus, the noise in data might dramatically change the center of this hyperquadric [3] if $F^{T}F$ is ill-conditioned. In the case of purely underdetermined linear systems, the situation is even worse, since due to its underdetermined character, the solution might completely absorb the noise in data without originating any data misfit.

The simplest example is a two-dimensional least-squares problem that consists of fitting the straight line y∗(x;m)=a0+a1x to a set of points (xk,yk). The least-squares problem consists of finding a set of parameters m=(a0,a1) so that the distance between the observed data vector yobs=(y1⋮ys) and the predicted values ypre(m)=(y1∗(m)⋮ys∗(m)) is minimized. The problem could be expressed in a matrix form as Fm=yobs, where the matrix F=[1ℝs x] depends in the abscissas of the data points x=(x1⋮xs).

Understanding and quantifying uncertainty in protein mutation is of utmost importance. It is analysis can be carried out directly by sampling equivalent model parameters, $m=\left(a_{0},a_{1}\right)$, whose predicted data fall within a specified error tolerance, $M_{tol}$.
(10)$$M_{tol}=\left\{m=\left(a_{0},a_{1}\right):\;\frac{\|y^{\mathrm{obs}}-y^{\mathrm{pre}}\left(m\right)\|{}_{2}}{\|y^{\mathrm{obs}}\|{}_{2}}<tol\right\}$$

Fernández-Martínez et al. demonstrated that the landscape of the cost function corresponds to a straight flat valley in case of linear problems, while in the nonlinear case, the cost functions have one or more curvilinear valleys of low misfits eventually connected by saddle points [2,3].

Figure 1 represents an uncertainty ellipse for different relative misfits and different sets of model parameters, $m=\left(a_{0},a_{1}\right)$. The main conclusion of the uncertainty analysis of this kind is that any classification problem has a deterministic nature. More specifically, it is possible to demonstrate that a simple robust method to sample the model parameters is to divide the problem into different random data bags and find the least-square solution of them.

In the case of nonlinear inverse problems, the region of uncertainty has a banana (or croissant) shape and might be composed of different elongated low-misfit basins. The noise also affects the region of equivalence in the case of nonlinear problems non-homogenously [4]. Moreover, the solutions of nonlinear inverse problems greatly depend on the prior information that is given to the optimization algorithm to find the minimum of the cost function C(m). Local optimization techniques use different regularization techniques to stabilize the inversion, in the process of finding a unique solution. Sampling methods have a different philosophy that consists of finding different models in $M_{tol}$ via intelligent search and do not need any regularization. The result of the sampling is not one unique model but a set of models that helps the modeler to make a robust decision. At the end, the solution of nonlinear systems via linearization implies that the noise on data impacts the solution of the linearized systems that have as an associated matrix the Jacobian of F calculated in the model where the linearization has taken place ($JF\left(m_{0}\right))$. In this case, the ill-conditioning of the Jacobian matrix varies along with the optimization process.

In this field, the probabilistic (or Bayesian) approach to the inverse problems is interesting, since it consists of determining the posterior distribution of the model parameters. This approach was formerly proposed by Albert Tarantola and Bernard Valette [5,6], In fact, the Bayes rule applied to inverse problems is:(11)$$P\left(\frac{m}{d^{obs}}\right)=\frac{P\left(m\right)\cdot P\left(\frac{d^{obs}}{m}\right)}{P\left(d^{obs}\right)}$$

The term P(m) is the prior distribution of the model parameters; $P\left(\frac{d^{obs}}{m}\right)$ is the likelihood and depends on the cost function $C\left(m\right)=\frac{\|F\left(m\right)-d^{obs}\|{}_{2}}{\|d^{obs}\|{}_{2}}$; and $P\left(d^{obs}\right)$ is the evidence. The posterior distribution of the model parameters $P\left(\frac{m}{d^{obs}}\right)$ is given by this probability factorization. A model is more probable if its likelihood is higher–lower C(m), and it is compatible with the prior information. In practice, the evidence, $P\left(d^{obs}\right)$ is a normalization constant that considers the probability of having observed $d^{obs}.$ In the case of nonlinear inverse problems, the posterior probability distribution does not admit analytical expression as in the case of linear problems, where the posterior distribution is Gaussian. Sampling methods are needed to approximate this posterior distribution. Nevertheless, this sampling procedure is hampered by the curse of dimensionality for parameter identification problems [7]. This result generalizes the previous analysis performed for isotropic spaces, where it is possible to prove that the probability of sampling inside a hypersphere is almost impossible for more than 10 dimensions.

### 3.2. Modeling Errors

In modeling, it is crucial to differentiate between noise and modeling errors. Modeling errors refer to an incomplete or inaccurate problem understanding; that is, the mathematical model F is partially incorrect, and $F^{true}=F+\delta F,$ where δF represents the modeling theory errors.

In the case where the model is linear, it is very easy to show that the modeling errors can be interpreted as modeling noise if we know an approximation $\tilde{m}$ of the model **m**:(12)$$F^{true}m=\left(F+\delta F\right)m=d^{obs}\to Fm=d^{obs}-\delta F.\tilde{m}$$

In the case of a nonlinear problem, similar reasoning can be made by using the Jacobian approximation of the nonlinear operator F(m), calculated in $m_{0,\;}JF\left(m_{0}\right)$, as follows:(13)$$F\left(m\right)=F\left(m_{0}\right)+JF\left(m_{0}\right)\left(m-m_{0}\right)+o\left(\|m-m_{0}\|{}_{2}\right).$$

Therefore, in the presence of modeling errors, the solution of the inverse problem that is found always differs from the true solution.

Noise and modeling errors, although different in genesis, impact the solution of the inverse problem and highlight the importance of performing a correct uncertainty analysis.

### 3.3. The Curse of Dimensionality

The curse of dimensionality refers to the impossibility of sampling high-dimensional spaces without any prior model for dimensions greater than 10. This phrase was first coined by Bellman in the context of optimization over a large number of variables [7]. In fact, the conditional probability of sampling a point inside a hypersphere knowing that the point is inside the hypercube that inscribes it is practically nil for more than 10 dimensions. In the case of inverse problems, the associated uncertainty spaces are anisotropic; that is, the uncertainty is not the same in all directions [8], and the number of effective dimensions to be sampled decreases to a maximum of five variables.

Therefore, model reduction methods are needed to handle the sampling problem in very high-dimensional spaces. Practitioners in many disciplines do sacrifice the analysis of uncertainty due to its complexity. Instead, very complex artificial neural networks are trained on many examples to learn the attributes and build the perfect classifier. This is the case with deep neural networks (DNNs) and convolutional neural networks (CNNs) [9,10]. In this case, the over parametrization of the model (number of model parameters) and the data spaces (number of examples) allow for tuning of the neural networks’ parameters. In fact, this kind of system can be written in short form:(14)CNN(m)=dobs,
where CNN is a vector field from ℝN to ℝs, where *N* is the number of parameters of the neural network and s is the number of data examples. The fact that the neural system is overdimensioned, both in the model parameters and in the examples, makes the Jacobian matrix JCNN(m0) for a given configuration m0 rank deficient. Therefore, there exist many CNN configurations that can learn and optimize the prediction problem. Nevertheless, the question remains if these architectures can be simplified, and which examples do we need to take compulsory into account to optimize the decisions.

## 4. Specific Problems

### 4.1. Phenotype Prediction in Drug Design

The first example that we want to examine is phenotype prediction in drug design. Phenotype prediction consists of identifying the set(s) of genes that influence the disease genesis and development, which are typically considered altered genetic pathways.

The main challenges for this analysis are the following:✓The lack of a conceptual model that relates the different genes/probes to the class prediction.✓The incomplete knowledge of genetic functions and genetic pathways, which introduces errors in the analysis.✓The noise in the genetic data and class assignment, which introduces errors in the altered pathways. Particularly, it has been found that the preprocessing techniques used in genomic data collection and treatment have a great impact [11,12].✓The highly underdetermined character of the phenotype inverse problem, since the number of monitored genetic probes is much greater than the number of observed samples. In fact, the genomic big data is never big because the patient dimension is always relatively small. Therefore, it is preferable to talk about little big data.

As a result of all these realities, the phenotype prediction problems exhibit a huge associated uncertainty space, which may lead to the ambiguous characterization of biological pathways (many tantamount genetic networks that predict the phenotype with similar accuracies) and of the associated therapeutic targets.

A straightforward conclusion of this brief analysis is that it is impossible to discover the genetic pathways involved by considering just the best high-discriminatory genetic signatures (with the lowest prediction error), since these genes might not recurrence when analyzing other independent datasets. The solution to this problem consists of performing the sampling of the uncertainty space in phenotype prediction and enforcing the hypothesis of biological invariance, as we explain later in this paper.

The lack of a conceptual model is solved by constructing a classifier between the group of genetic signatures g and the group of classes C={1, 2} in which the phenotype is divided:(15)L∗(g):g∈ℝs→C={1, 2}.

The phenotype distribution C={1, 2} might correspond to disease and control samples to understand the altered genetic pathways and also to perform treatment optimization and understand the mechanisms of action of a drug and minimizing side effects, separating responders from nonresponders [13]. The uncertainty space of $L^{*}\left(g\right)$ is in this case:(16)$$M_{tol}=\left\{g:\;\|L^{*}\left(g\right)-c^{\mathrm{obs}}\|{}_{p}<E_{tol}\right\}$$
which is composed of the discriminatory genetic signatures that predict the observed class $c^{\mathrm{obs}}$ within the same error bound $E_{tol}$. Here, $\|L^{*}\left(g\right)-c^{\mathrm{obs}}\|{}_{p}$ represents the optimization of the cost function $O\left(g\right)=\|L^{*}\left(g\right)-c^{\mathrm{obs}}\|{}_{1},$ which comprises the observed classes $\;(c^{\mathrm{obs}})$ of a group of samples in the training dataset **T**, and the analogous set of predictions $L^{*}\left(g\right)$, by the genetic signature g and the classifier $L^{*}\left(g\right)$. Expressed as an inverse problem, $c^{\mathrm{obs}}$ is the observed data, $L^{*}\left(g\right)$ is the phenotype prediction, and $\|L^{*}\left(g\right)-c^{\mathrm{obs}}\|{}_{1}$ describes the prediction error, which is the distance between the observed and predicted classes, corresponding to the number of failed samples predicted by the classifier.

### 4.2. The Uncertainty Space in Phenotype Prediction

Phenotype prediction problems are deeply underdetermined, since the genetic information is always much larger than the number of observed samples. The uncertainty space related to the classifier $L^{*}$, $M_{tol}=\left\{g:\;O\left(g\right)<E_{tol}\right\}$, is made up of sets of highly predictive networks with analogous predictive accuracy, i.e., sets of genes g that classify the samples with a prediction error O(g) lower than $E_{tol}$. These groups of genes reside in one or several flat curvilinear valleys of the cost function topography, O(g) [2,14]. Therefore, due to the highly underdetermined character of this kind of prediction problem, the related uncertainty space has an immense dimension, and the characterization of the biological pathways that could be involved is very ambiguous since there might exist many equivalent genetic networks that predict the phenotype with similar accuracy. The biomedical robot is defined as a set of approaches from applied mathematics and computer science, capable of dynamically learning from high-dimensional data, performing predictions with uncertainty assessment, and generating valid, working hypotheses [15]. Artificial intelligence (AI) methodologies, undoubtedly, have an important role in this. Particularly, the legibility and the comprehensibility of these algorithms are of paramount importance to understanding why the drug design is successful. Therefore, if possible, the use of black-box models should be avoided.

The concept resides in sampling the uncertainty space to discover the altered genetic pathways and use them in the drug design and drug repositioning processes. Cernea et al. and Fernández-Martínez et al. [16,17,18] presented different sampling techniques related to phenotype prediction. This methodology was applied to the analysis of defective genetic pathways in triple breast cancer, comparing the results that were obtained with Bayesian networks, which are directed acyclic graphs used to encode the conditional probability distribution between variables. Bayesian networks help to sample the posterior distribution of the genetic signatures related to the phenotype prediction, $P\left(g/c^{\mathrm{obs}}\right)$. According to Bayes’ rule for this problem:(17)$$P\left(g/c^{\mathrm{obs}}\right)∼P\left(g\right)\;P\left(c^{\mathrm{obs}}/g\right)$$

In this expression, P(g) is the prior distribution for sampling the genetic signatures, and $P\left(c^{\mathrm{obs}}/g\right)$ is the likelihood of the genetic signature g, which depends on its predictive accuracy O(g). These algorithms have been recently applied to perform the robust sampling of the altered pathways in different diseases: Parkinson’s, Alzheimer’s, multiple sclerosis, multiple myeloma, and triple-negative cancer [19,20,21,22,23].

The main source of noise in phenotype prediction problems is due to the presence of noise in data and also in the class assignment [11]. The noise in data comes for instance when measuring gene expression in the transcriptome, either by RNA-seq or microarray techniques. It follows that these data are typically contaminated by measurement noise that introduces errors in the genetic measure. Moreover, it has been proven that the preprocessing treatments of the raw data performed for the microarrays tend to introduce errors in the altered pathways [12]. Furthermore, this study shows that the noise in the class assignment has a more prominent impact than noise in the expression data.

Summarizing, the uncertainty in phenotype prediction has the following sources:○Lack of a conceptual model explaining the disease: the lack of this model might provoke the discovery of spurious relationships.○Highly underdetermined character: the number of patients is very reduced compared to the number of possible causes.○Noise in data (genetic expressions and class vectors).○The question/problem statement itself might be partially erroneous.

### 4.3. Drug Discovery and Drug Repositioning/Repurposing

Drug development is a capital-intensive process. The discovery of new drugs and their development to reach the market in the pharmaceutical industry is an exceedingly expensive, long, and stringent process, with an average time of 10 to 16 years. In addition, the high attrition rates in clinical trials are mainly due to the lack of effectiveness and possible toxicities. The analysis and comprehension of mechanisms of action (MoA) of drugs and the prediction of toxicities are two major challenges of drug development [24]. Accordingly, the policy of one disease–one target–one drug should be reconsidered, and drug repositioning seems to be one of the possible answers. Drug repositioning aims at finding the drugs that are already approved by FDA that can equilibrate the altered genetic pathways and achieve homeostasis. Once the altered genetic pathways are robustly sampled, the next step should be performed: optimum compound selection. The main goal of this procedure is to dramatically increase the approval rate in drug design and the repositioning of existing drugs. This is of paramount importance in the case of rare diseases for understanding the disease mechanisms in the search for orphan drugs.

To optimally delineate the targets and compounds, we should use genomic, proteomic, and metabolomics information. The ongoing view in drug design is that the mechanisms of action should be able to reestablish homeostasis. For such purposes, the drug and the disease are treated likewise; that is, the drug should hit the targets that are perturbed by the diseases, trying to reestablish the equilibrium.

A conceptual scheme of a predictor–corrector methodology for drug design has been proposed in [25]. Figure 2 shows the scheme of this predictor–corrector methodology, which is applied iteratively between the preclinical models and clinical trials to optimize their FDA approval. Two critical points of this methodology are:○Performing a correct selection of the potential compounds in the preclinical trials to maximize efficacy and minimize toxicity.○Trying to predict the outcomes of clinical trials via preclinical data and AI methods. This implies an adequate data acquisition pipeline.

The uncertainty in drug discovery has the following sources:○The sampling of the altered pathways, as seen in the phenotype prediction problem.○Lack of a conceptual model relating the expression of the altered genes to the list of selected potential compounds, otherwise considered the hypothesis of achieving homeostasis via the effect of these compounds. Moreover, the CMAP data (methodology) comes from in vitro experiments and the altered pathways should be rebalanced for humans.○Incomplete coverage of CMAP data: the perturbed genes, the compounds, the tissues, and the doses.○Lastly, the extrapolation from preclinical analysis (in vivo and in vitro) to clinical trials (humans) with respect to the mechanisms of action (MOA) and the generation of toxicities (side effects) is not obvious.

Connectivity Map (CMAP) is a library containing over 1.5 M gene expression profiles from ~5000 small-molecule compounds to ~3000 genetic reagents, tested in multiple cell types, developed at the Broad Institute (https://www.broadinstitute.org/connectivity-map-cmap (accessed on 1 December 2021)).

### 4.4. De Novo Design: Sampling the Chemical Space

De novo drug design consists of finding/generating novel molecules with specific pharmacological properties to reverse a disease. These methods have become highly popular in the last two decades due to a better understanding of biological systems and the advances in computing techniques, mainly deep learning approaches.

The main big challenge of de novo drug design is exploring the chemical space due to its huge dimension. It is estimated that the number of possible drug-like compounds in chemical space may be up to 10^23^–10^60^. Moreover, only $10^{8}$ have been discovered in nature or synthesized [26,27].

The created mathematical models are mathematical representations that link the structure of a molecule with its effects. Therefore, one important problem to be solved concerns the molecules parameterization, to create models between the molecules’ representations and their properties, that should be annotated in the training dataset.

In the last few years, the concept of artificial intelligence (or creativity) has been implemented using deep generative models that have allowed the creation of ever more autonomous and efficient systems in contexts of art, music, and literature. Inspired by these improvements, now, these advances and techniques have been applied to molecular generation and optimization, reducing costs, time, and resources spent on synthesizing incorrect leads in laboratories or research centers. Recently, several emerging generative models based on deep learning have proved to be viable and efficient solutions, being able to optimize multiple pharmaceutically important parameters simultaneously for de novo drug design, resulting in better exploration of chemical space and helping in finding novel chemical structures for drug development.

The use of deep learning (DL) approaches, that is, sets of algorithms that use artificial neural networks (ANNs), with multiple layers of nonlinear processing units for modeling high-level abstractions contained in the input data is very appealing in drug design. Considered learning methods, these are a kind of black-box models that relate abstract representations of the molecules with their observed properties. One practical example is the following: let us imagine that based on in vitro experiments, we have at our disposal a set of drugs that are able, in a tissue at a given dose, to reduce the expression of a given gene that is clearly overexpressed in a disease. In this case, the deep learning model will try to relate the structure of the molecule and other measurable and important properties to its effect in the expression of this gene, that is, to mimic or create a forward model to predict this effect. It is important to remark that in this case, as in many others in the biomedical field, these forward models are unknown. Moreover, one of the main difficulties in this DL design is that the number of examples at disposal is typically very low. Nevertheless, if this design was successful, we will be able to predict the effect of this gene on a compound that is not in the training set and maybe to understand, which are the important parts of its structure that play a crucial role in this effect.

Concerning the use of deep generative modeling of molecules, one of the preliminary works was the autoencoder adopted by Gomez-Bombarelli et al. [28]. These authors used the technique based on a gradient-based optimization of molecular properties in a continuous and differentiable latent space using SMILES, a string-based representation derived from molecular graphs. It is very important to correctly understand the AI language that is involved in this terminology. The latent space is the reduced dimensional space that is used to represent the molecular structure of the compounds and the autoencoders are nonlinear dimensionality reduction algorithms to perform this task.

Since this seminal work, several advancements have been developed, such as recurrent neural networks (RNNs), autoencoders (AEs), and generative adversarial networks (GANs), among others. Recurrent neural networks are a kind of artificial neural network (ANN), which, unlike feed-forward networks, include feedback components that permit signals from one layer to go back to a previous layer, resulting in an output influence not just by weights applied on inputs, but also by a hidden state vector representing the context based on prior input/output. This interesting change from basic neural networks is applied by connections between units to generate an internal state of the network that can display dynamic temporal behavior. This is possible by the most commonly used units such as long short-term memory (LSTM) [29] or a newer, more computationally efficient variant called the gated recurrent unit (GRU) [30].

Autoencoders (AEs) are a class of unsupervised neural networks, mostly used for feature extraction, denoising applications, and dimensionality reduction, in which the output is set to be equal to the input employing a back-propagation algorithm for the training [31]. Having this last method as a base, some other advancements have been developed such as variational autoencoders (VAEs), which make use of the Bayes rule to sample the latent space in a probabilistic way. These variational autoencoders have shown better performance, since they are able to learn more robust representations [32]. The closest idea to autoencoders is dimensionality reduction via principal component analysis (PCA), which uses a linear method for the expansion based on orthogonal matrices. Therefore, passing from the original space to the reduced space is very fast.

Generative adversarial networks (GANs) were proposed by Goodfellow et al. in 2014 [33]. They belong to the set of generative models that are aimed at the generation of new proposals. In a GAN setup, two differentiable units represented by neural networks are involved: a generator and a discriminator. The generator network directly produces samples and tries to maximize the probability of making mistakes by the discriminator in characterizing inputs as real, and the discriminator, whose job is to distinguish whether the molecule it is looking at is generated by the generative model or comes from the training data.

It is important to understand that GAN is a method to create new compounds. In fact, the problem can be stated as follows: given a set of models with given measurable properties (positive and negative), the question is how to generate new models from this training dataset maximizing a given property. Therefore, solving the GAN architecture needs the solution of complex optimization problems. Coming back to the last example of the overexpressed gene, in this case, the GANS will be used for instance to generate a new compound that reduces the expression of this gene and minimizes for instance the outcome of side effects. GANs should be viewed as smart perturbation methods.

In the last few years, more common deep learning techniques have been described, and some other improvements have been studied, more specifically, in the de novo drug design field. First, based on the autoencoder by Gomez-Bombarelli et al., an interesting technique combines a variational autoencoder with a multilayer perceptron (MLP) to create new molecules with desired properties. The network consisted of an encoder that converts SMILES strings into continuous vectors in latent space and a decoder that can turn these vectors back to a discrete SMILES string, and a multilayer perceptron that predicts the properties of the molecules [29]. This algorithm has as the main limitation the generation of many cases of invalid SMILES resulting in unfeasible chemical structures. A successful way to overcome this limitation was a grammar VAE developed by Kusner et al. [34] and Dai et al. [35] that explicitly adds syntactic and semantic constraints to SMILES strings using context-free and attribute grammar.

Recently, there has been an increasing interest in using RNN for the de novo design of molecules. Segler et al. introduced the notion of transfer and reinforcement learning for generating focused molecule libraries using RNNs on SMILES [36]. The generative LSTM-based model pretrained on a general corpus of molecules was either fine-tuned on a small number of known actives or coupled to an external scoring function for creating novel structures with desirable activity against a specific biological target. Another interesting strategy using what the author named SMILES enumeration was proposed by Bjerrum [37], defining this as a single-line text uniquely representing one molecule, using it as the raw input to the LSTM cell-based neural network to build predictive models without the requirement for creating molecular descriptors. The results showed that SMILES enumeration produces statistically more robust QSAR models when predicting single SMILES, but even more when taking the average prediction of using enumerated SMILES for the same molecule.

Some other DL achievements have been reached in the fields of molecular property and activity prediction, showing better performance compared with older machine learning techniques for different applications, such as the prediction of biological activity and physicochemical parameters. One of the first developed examples in drug discovery dates to 2012, when a multitask deep feed-forward algorithm showed an improvement in the prediction of drug properties and activities of about 15% in relative accuracy over the pharmaceutical proprietary system [38]. Later, other groups proved that massively multitask DL architectures perform better than single-task and random forest models in property prediction [39,40]. Another example is the application of the classification of several drugs into therapeutic categories solely based on their transcriptional profiles in different cell lines combined with pathway information [41], training DL algorithms on large transcriptional response data sets. A DL approach named undirected graph recursive neural networks (UG-RNN) [42] was used to effectively predict the water solubility of compounds.

Although significant progress on DL methods and applications has been made in the areas of de novo drug design and drug discovery, some outstanding challenges remain, such as the dependency on using experimental data for training and validation. However, in recent years, some advancements have been developed to overcome this and other difficulties in the fields of transfer learning [43], one-shot learning [44] and reinforcement learning [45], or deep generative models [46]. Furthermore, complementary research into the generation of new descriptors, such as including fingerprint-based systems [47], as well as the development of supplementary criteria for a correct representation of a chemical system, will be essential. The availability of flexible and adaptable DL models can accelerate future developments via learned features and theory-informed models.

The main causes of uncertainty in de novo drug design are:○The huge dimension of the chemical space.○The lack of a conceptual model relating the chemical structure of one compound to its effect in transcriptomics. The idea is to train AI algorithms to learn the chemical space.○Highly underdetermined character of the problem since there are not enough examples to train these neural networks.○The parameterization of the compound structure is also an additional source of ambiguity.○Noise in experimental data (mainly genetics).○The question/problem statement itself might be partially erroneous: can the structure of the compound be related to its effect?

**DrugBank** is a comprehensive, freely accessible, online database containing information on drugs and drug targets, available at www.drugbank.com (accessed on 1 December 2021).

### 4.5. Uncertainty in the Prediction of Protein Mutations

Identifying amino-acid substitutions that have an impact on protein function and their implications in disease is one the forefront challenges in proteomics, metabolomics, and medical genomics [48].

One single individual could experience between 24,000 and 40,000 amino-acid substitutions. Among these variants, the most important ones are single-nucleotide variants (SNVs) [49,50,51], which are directly related to different diseases. However, the increasing amount of gathered data has not affected the understanding of these disease mechanisms due to the problem’s complexity. The accuracy of these methods remains limited, since high false-positive rates are observed [52]. On top of this, the majority of tools are focused on achieving high accuracy without understanding the links between the classification attributes and the final predicted result [53,54,55]. Finally, another major drawback of these supervised machine learning methods is that decision rules are based on a set of overlapped training datasets, yielding higher performance estimates [56,57]. Consequently, it is very important to train, test, and validate the performance of these tools on completely independent datasets [58]. This problem is further enhanced by the lack of variability in training datasets, yielding biased prediction [59].

Predicting the effect of single amino-acid polymorphism (SAP) is a problem of performing a robust classification. Predicting the effect of mutations is a classification problem. We can understand the prediction of protein mutations as a generalized regression problem taking place between the protein discriminatory attributes that characterize the mutation effect and the mutation classes, either neutral or deleterious. Consequently, a key step in developing a machine learning framework to predict the effect of mutations is to forecast the effect via consensus, defining and implementing an algorithm that optimally selects the partial results to be combined in a consensus classifier that selects the final prediction.

The main causes of uncertainty in the prediction of protein mutations are:○The lack of a conceptual model relating the mutation to its effect. The idea is to train AI algorithms to learn the chemical space.○Highly underdetermined character of this problem since there are not enough examples to train these neural networks.○The unknown effect of simultaneous mutations.○The extrapolation of the effect of one mutation in one protein to other proteins.○Noise in experimental data (mainly proteomics).○The question/problem statement itself might be partially erroneous: can the structure of the compound be related to its effect?

ClinVar is a public archive on the relationships between human variations and phenotypes available at https://www.ncbi.nlm.nih.gov/clinvar/ (accessed on 1 December 2021).

### 4.6. Different Approaches to Decrease Uncertainty

A machine learning problem is said to be well-posed if a solution to it exists, if that solution is unique, and if that solution depends on the data but it is not sensitive to reasonably small changes in the data. These three causes of ill-posedness are important with respect to the uncertainty problem.

First, in the case where the solution does not exist, it implies that the question is somehow unfeasible, and the problem is recast in the least-squares sense. In this case, we should consider that our learning model is partially incorrect. If the learning misfit is very high (higher than 25%), the result of the modeling cannot be expected to be successful.

In the case where the solution is not unique, the methodology should be able to sample all the plausible scenarios. Sampling methods in nonlinear problems are always hampered by the curse of dimensionality. Therefore, feature selection and model reduction methods are deeply needed.

Stability is not an issue when sampling methods are used, because they do not compute the inverse operator of the forward problem. Nevertheless, the computation resources needed to perform the forward predictions might hamper the use of sampling methodologies. In this case, fast proxy methods (surrogate methods) are needed to fasten these computations.

With respect to data, there are two main issues: noise in data and incomplete coverage of the data space. Both have very bad consequences. In the first case, we do not recommend using any kind of filters to smooth the data, because their effect could be worse than the problem they tried to avoid. Our methodologies should be robust to the effect of noise in data. For instance, the data coverage in the phenotype prediction problem will always be incomplete, since the number of samples (patients) will always be much lower than the number of possible genes that might account for the disease development. Therefore, robust feature selection methods are needed to reduce the dimension of the possible causes and find the different alphabets. The word alphabets means that there will be multiple scenarios/solutions that are equally consistent with the data. It follows that increasing the number of data that is available helps, but in genetic studies, we will always have the case of little big data on the side of patients, compared to the number of genetic variables.

Finally, the fact that the theory is unknown or incomplete is something that we must deal with. For that purpose, the artificial intelligence methods that are built should try to be transparent, that is, be able to generate hypotheses. The term biomedical robots [15] accounts for a system that can dynamically learn from complex high-dimensional data, generate a working hypothesis in a transparent way, and perform predictions with its corresponding uncertainty that is able to sample equally plausible scenarios.

## Figures and Tables

**Figure 1 ijms-23-12975-f001:**
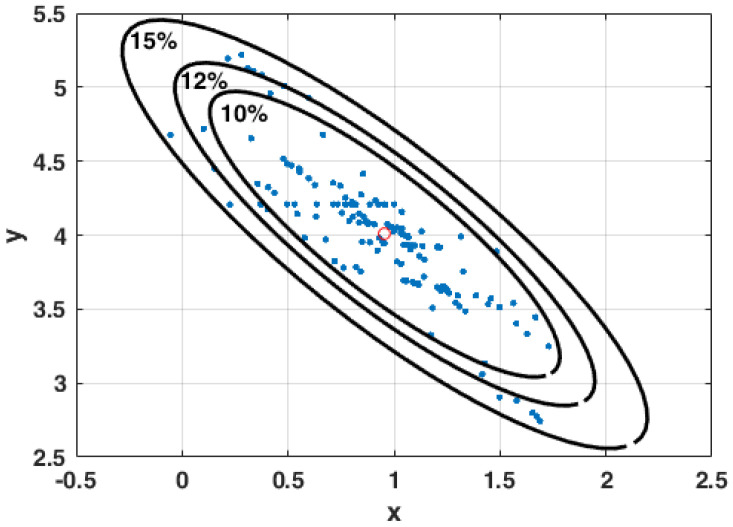
Example of a linear regression model. Ellipse of uncertainty for different relative misfits ranging from 10% to 15% and different sets of model parameters found in the different bagging experiments.

**Figure 2 ijms-23-12975-f002:**
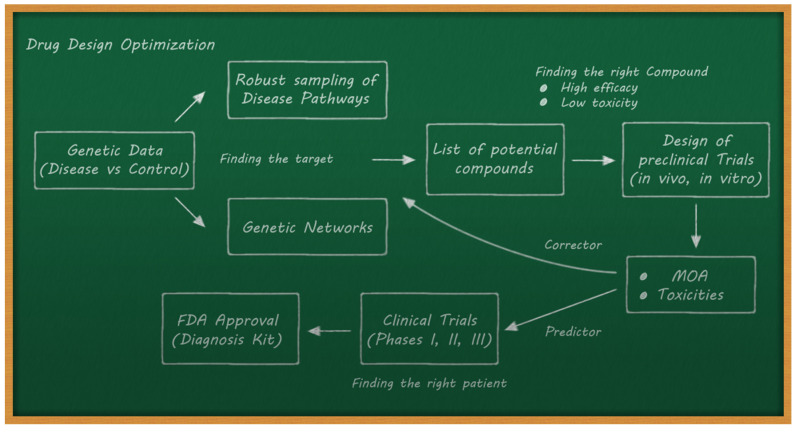
A possible predictor–corrector methodology for drug design.

## Data Availability

Not Applicable.

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
