# Peer review of "Addressing Noise and Estimating Uncertainty in Biomedical Data through the Exploration of Chemical Space"

_ijms, 2022, doi:10.3390/ijms232112975_

Round 1

Reviewer 1 Report

The manuscript "adressing noise and estimating uncertainty ..." tackles an important topic of in-silico predictions. Noise and resulting uncertainty are important considerations which need to be considered in predictions for directing experimental efforts by in-silico methods. The authors convey this message with quite some mathematical overhead. Mathematical formalism is introduced, but when it comes to concrete applications in the second part of the manuscript, the formalism is not used anymore but quite general statements about noise and uncertainty are used, which are not supported by the formalism introduced. One example for that around lines 495 where the conclusion about deterministic nature is simply stated but not derived from the formalism nor any reference cited. Statements about property prediction are very general (lines 430ff) and only few references regarding the significant work about uncertainty prediction in machine learning, just to name conformal prediction. This imbalance between framework set by the mathematical formalism and details in the second part is a motif throughout the manuscript. Authors should take a decision, if they would like to give the overview (which needs significantly more citations and examples) or the mathematically derived framework (which requires a more thorough presentation).

The term falsify appears to be used in the meaning of introducing errors. I recommend to change that in light of the thinking of falsification of theories introduced by K. Popper.

Author Response

Reply to the Reviewer Report is Attached

Reviewer 2 Report

The manuscript is devoted to the review of noise and uncertainty problems in biomedical data through the exploration of chemical space. The work is interesting and worth attention. However, I give several comments below.

1. The literature is relevant, but most of the sources are quite old. It is recommended revising the references, excluding the out of date sources, and remaining the sources within up to 5 years old.

2. Some parts of the review are less connected with the main topic. They are 3 and 3. It is recommended to make closer connections between those parts and the main problems of the study.

4. It is recommended to restructure the work according to the cases that are presented in the Introduction in lines 81-98. Subsections should reflect answers to those cases.

5. Lines 376-404 present a general description of ANN's without strong relation to the context. The authors should revise paragraphs to make closer connections with the main problems.

6.The authors should consider adding examples and description of the biomedical data sets along with descriptions of their noise and uncertainty. The links to such datasets are appreciated.

Reviewer 3 Report

Thank you very much to a very nice review paper.

- The abstract needs to be rewritten as it is too generic and does not fully reflect your contribution in the paper. We know that artificial intelligence can provide solutions to uncertainties in data, but how? It would be nice if the abstract was more specific and tailored to the problem you are addressing.

- In Highlighs Lines 25 - 36. I think you may have, perhaps, overclaimed too much. Please tone them down. "The biggest uncertainty is introduced by the question that we try to ellucidate." What do you mean by that? "Sampling is the only way to address both issues" Really!? I do not think you are right on this point. You need to back this up with arguments, and the arguments you make in your paper are not really convincing. 

- Umm. I think there is a problem with your sections. Introduction -> 1. The concept ... -> Methodology -> 2. Inverse problems -> so on.. Please choose one style and update them accordingly as it is very much confused. Please change.

- Please check grammars, typos and your writing again. E.g. Lines 204 - 205: "In this case the over-parameterization of the model (number of model parameters) and the data spaces (number of examples) allow for the tuning of the neural networks" sounds incomplete and grammatically incorrect. Line 525 "diffenret" !?

- The paragraph following an equation should not be indented.

- It would be nice to add a table summarising noises and uncertainties in each specific problem you have outlined and methods to address the issue.

- You have talked a lot about the simple topics that people already know. From the title of the paper, I was expecting to read methods/strategies for dealing with noise and uncertainty in biomedical data, but I do not see much of that in the paper. Perhaps you should change your title to better reflect the content, or add more content that relates to the title. When you say "The only way to address both issues (noise in data and uncertainty) it would be nice if you added more hard content about sampling techniques (the way they are implemented, advantages and disadvantages, etc.).

- Basically, you need to restructure your manuscript and connect it with some more content to make it more complete. Please also address grammartical mistakes.

Round 2

Reviewer 1 Report

The manuscript got a bit clearer and more structured esp. in the second part. 

Author Response

Thank you very much!

Reviewer 2 Report

I recommend accepting this manuscript.

Author Response

Thank you very much!

Reviewer 3 Report

The authors have addressed all the concerns I have raised. However, please consider changing the title to make it more concise..

Author Response

Thank you very much! We have shortened the title as requested.